# ABC model for cost estimation of custom implants by Additive Manufacturing

**Zaineb Hameed Neamah** [1]*, **Luma A. H. Al-Kindi**[1], **Ghassan Al-Kindi**[2]

**1** Department of Production Engineering and Metallurgy, University of Technology, Baghdad, Iraq,
**2** Department of Mechanical Engineering, Sohar University, Sohar, Oman

* pme.20.38@grad.uotechnology.edu.iq

## Abstract

Computer-aided design (CAD) models can now be directly converted into products and structures. One technique to realize such approach is through Additive Manufacturing (AM). AM is relatively new manufacturing technology in which products are manufactured by layering various materials like rubber, metal, ceramic, composites, and polymers. However, the use of this technology requires consideration of its associated cost to ensure its competitiveness. In this paper, a simplified mathematical cost model is suggested. The model considers the main components of costs. The model formula utilizes expenses related to the pre-processing, main processing, and the post-processing operations. To validate the model, it is tested to estimate the cost of medical implants manufacturing using AM technique. In many cases, medical implants require unique or dedicated design for each patient. Hence cost estimation will help to assess and estimate the required financial resources for such operations. A case study is provided in this paper to estimate the manufacturing cost of a finger's phalanges bone, with metal implant using AM technique. The developed model may be described as Activity Based Costing (ABC). The model is introduced to estimate the cost of parts produced using AM technique. Although the model is developed to suit custom implant manufacturing using AM technique, its use may also be adapted to suit the manufacturing of many other parts and products. The developed model is aiming to achieve several tasks namely assigning cost drivers to each activity, estimating the cost of individual actions, allocating overhead expenses, calculating the overall production cost, and establishing an acceptable selling price. It assists companies in computing the cost of custom implants for customers, enhancing the accuracy of production cost estimates, and ultimately boosting profitability.

## 1. Introduction

Now a days, Additive Manufacturing (AM), often known as 3D printing, is a technology that is making significant progress in customizing medical implants for individuals. It enables the creation of complex geometries impossible with traditional production methods [1, 2]. Therefore, implants that need to fit into unusually shaped areas, such as the skull or pelvis, can benefit greatly from this innovation. Polymers, metals, and ceramics are among the materials that can be used in the production of implants through AM. Hence, selecting the best implant

**Competing interests:** The authors have declared that no competing interests exist.

material becomes more accessible based on its function and the patient's needs [3, 4]. The increasing cost is directly attributable to the difficulty and time required to create custom orthopedic implants using AM [5, 6]. Hence, it is necessary to find ways to estimate and reduce the time and cost required to perform custom bone implants using AM, while maintaining quality and safety standards for patients. Estimating costs helps for decision-making as product designs and production processes change. Additionally, providing consumers with a clear cost estimate is advantageous [7, 8]. The resources used to make a product affect its price which are machinery, materials, labor, and equipment all fall into this category. Therefore, the costs of these components must be considered to arrive at the final cost of the medicinal product. The primary motivation for this research is to determine the approximate cost of a medical implant manufactured by AM. We plan to use a cost calculation approach, for the model allowing for an allocation of indirect costs, across the three stages of processing [9, 10]. The method utilized for estimating expenses referred to as the Activity Based Costing (ABC) model dissects the costs associated with each step, in producing a product. By segmenting the process into activities and assigning a cost to each one the ABC model can accurately calculate the production cost of personalized implants manufactured by AM [11]. The steps for developing an ABC model for cost estimation include Step 1. Gathering data on the activities involved in manufacturing custom implants materials used and the costs and duration of each activity. Production records, time sheets and cost accounting reports serve as sources of this information. Step 2. Identifying cost drivers to pinpoint factors that significantly impact the cost of each activity; this may necessitate input from the production team. Step 3. Computing activity costs using these identified cost drivers. This could entail determining the cost per unit of a driver (labor hours cost per hour) or projecting necessary resources for each activity and multiplying it by the resources expense. Step 4. Apportioning costs by estimating what percentage of expenses are linked to each activity. Step 5. Calculating the cost by summing up individual activity costs along, with expenses.

Despite the significant importance of 3D printing in the medical field, only a few research papers address the cost estimation of custom implants by AM. In [6], the authors develop a cost model while adhering to the ABC estimation principle. They factor in the expenses of three procedures in their approach. During the pre-processing step, the machine operator uses the setup software. Processing the item involves printing it in successive layers using an AM machine. Finally, in post-processing, the printed parts are retrieved by the machine operator, followed by preparing the AM system for the subsequent production run. In [7], the authors suggest an approach to estimate the cost of metal AM systems. They used a process-based method to construct their cost model. In developing it, they considered materials, labor, energy, and equipment prices. By constructing high-stress parts with lower power values to achieve greater yield strength and increasing power elsewhere to minimize the number of passes and construction time, they demonstrated that more cost-effective outcomes could be expected with the same amount of material. However, the suggested approach did not account for administrative expenses. This paper will utilize the ABC model to accurately estimate the costs of producing custom implants via AM, considering the various activities involved and the variables affecting these costs. Ensuring that implants are appropriately costed can help determine their accessibility and appropriate pricing for individuals in need [12, 13].

## 2. The methodology of total cost model calculations

In this study, the production of AM is divided into three distinct activities: pre-processing, processing, and post-processing. These activities encompass various cost elements, as illustrated in Fig 1. For that reason, a mathematical approach is proposed to facilitate the

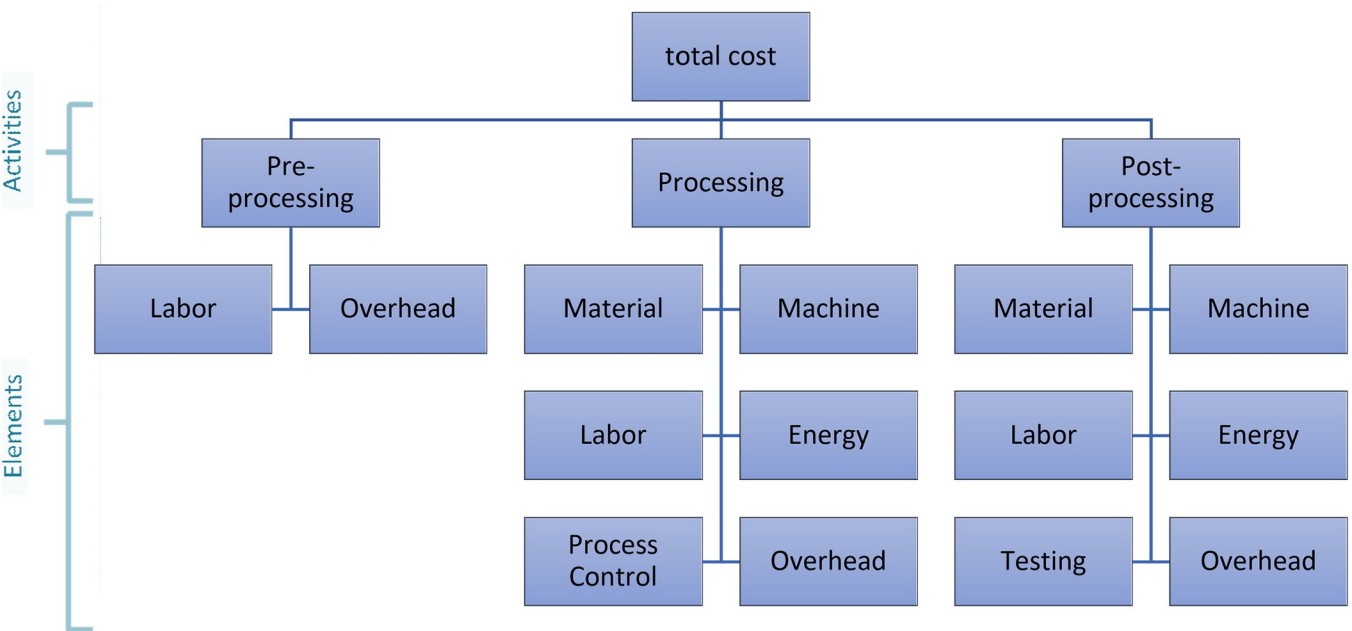

**Fig 1. Three main activities and their associated cost components [1].**

computation of the overall cost associated with the production of medical implants using the AM technique [14–16]. The cost model is developed by considering all these activities and their respective elements. Given that certain AM systems have the capability to construct multiple parts ($N$) within a single build, the total cost per part ($Total_{Cost}$) is represented as:

$$Total_{Cost} = \frac{1}{N} * \left( C_{prep} + C_{processing} + C_{post} \right) \tag{1}$$

where:
$C_{prep}$ = Pre-processing activity cost ($)
$C_{processing}$ = Processing activity cost ($)
$C_{post}$ = Post-processing activity cost ($)

## a. Pre-processing activity

The first activity in AM technology is the pre-processing one which involves all the necessary preparation works before building. The associated costs with this activity include labor and overhead costs. Specifically, the pre-processing cost ($C_{Prep}$) is calculated as follows; derived from the sum of labor cost ($C_{L_1}$) and overhead cost ($C_{OH_1}$), as outlined in Eq (2)

$$C_{Prep} = C_{L_1} + C_{OH_1} \tag{2}$$

**Labor cost.** During the pre-processing stage, a series of tasks must be completed by an operator. The commencement of the pre-processing activity requires the operator to possess a CAD file, which is assumed to be free of errors. Subsequently, the operator proceeds to convert this CAD file into an STL file, adjust the build orientation, generate support structures, slice the build, and determine the layer thickness. After the file is prepared, the file is transferred to

the AM system to begin printing. Eq 3 shows the calculation of the labor cost by multiplying the operator's hourly rate ($O_1$) by the time spent preparing the build ($T_{prep}$), encapsulated in the formula:

$$C_{L_1} = O_1 * T_{prep} \tag{3}$$

**Overhead cost.** The overhead in the pre-processing stage of AM encompasses costs not directly linked to production, such as administrative and manufacturing overheads. These include expenses like building rent, maintenance, depreciation, utilities, and licensing fees. Overhead costs, both administrative and manufacturing, are allocated based on the time spent in pre-processing, calculated by multiplying the overhead rate ($H_1$) by the preparation time ($T_{prep}$):

$$C_{OH_1} = H_1 * T_{prep} \tag{4}$$

This equation ensures a comprehensive account of all indirect costs associated with preparing for the manufacturing process.

## b. Processing activity

After all preparation actions are complete, the next activity begins. With this activity, the AM system starts to build the part. Cost components of processing activity include material, machine, labor, energy, process control, and overhead costs. Accordingly, the processing cost can be calculated as follows:

$$C_{processing} = C_{MAT} + C_{Mach} + C_{L_2} + C_E + C_{OH_2} \tag{5}$$

where:
$C_{MAT}$ = Material cost ($)
$C_{Mach}$ = Machine cost ($)
$C_{L_2}$ = Labor cost ($)
$C_E$ = Energy cost ($)
$C_{OH_2}$ = Overhead cost ($)

**Material cost.** The material cost for AM processing denoted as $C_{MAT}$, is calculated when different materials are used for the build and support structures using the equation:

$$C_{MAT} = (W_1 * CM_1) + (W_2 * CM_2) \tag{6}$$

where:
$W_1$ = Weight of the build material (kg)
$CM_1$ = Cost rate of the build material ($/kg)
$W_2$ = Weight of support structure material (kg)
$CM_2$ = Cost rate of support structure material ($/kg)

This equation accounts for the material costs required for creating both the primary structure and any necessary supports during the manufacturing process. When the same material is used for both the main build and support structures in AM, the material cost calculation simplifies. The cost is determined by the combined weight of the build and support material, multiplied by a single material cost rate, as depicted in Eq (7):

$$C_{MAT} = (W_1 + W_2) * CM_1 \tag{7}$$

**Machine cost.** The cost associated with the machine during the processing activity includes the initial investment, maintenance, and potential salvage value. It also factors in the time for both setup and actual building, acknowledging that the AM machine is dedicated entirely to the project during these phases. The formula for calculating machine cost is represented as:

$$C_{\text{Mach}} = \left( \frac{C_I + M - S}{T} \right) \times T_{bs} \tag{8}$$

where:

$C_I$ = Machine purchase value ($)
$M$ = Machine maintenance cost ($)
$S$ = Salvage value ($)
$T$ = The useful life of the AM machine (h)
$T_{bs}$ = Time required to setup and build (h)

This calculation provides a comprehensive view of the machine's financial impact on the processing activity.

**Labor cost.** The labor costs for different stages in AM, such as pre-processing and processing, vary due to the distinct tasks and skill levels required. For activities where operators are needed to initiate the system, manage materials, or monitor the build, the labor cost ($C_{L_2}$) is derived as:

$$C_{L_2} = O_2 * T_{processing} \tag{9}$$

where:

$O_2$ = Operator cost rate ($/h)
$T_{processing}$ = Time required for doing tasks (h)

This approach ensures accurate accounting of labor costs based on the specific contributions of operators during the production process.

**Process control cost.** Process control during the processing phase of AM is essential for ensuring part quality. This involves activities that can be either automated through devices integrated into the AM system or manually conducted by a quality control technician. The cost associated with automated process control is included in the machine cost as shown in Eq (8). When process control is performed manually, the cost is included in the labor cost as shown in Eq (9). This differentiation helps ensure that costs associated with quality assurance processes are appropriately allocated.

**Energy cost.** The energy cost during the processing activity $C_E$, is calculated as:

$$C_E = E * T_b * E_R \tag{10}$$

where:

$E$ = The AM System's Energy Use (Kw)
$T_b$ = Time to Build (h)
$E_R$ = Cost of Energy Rate ($/Kwh)

This formula computes the energy consumed by the AM system specifically for part production, and does not account other types of energy consumption grouped under overhead costs.

**Overhead cost.** The overhead costs during the processing phase, administrative services, lighting, etc. are proportioned based on the setup and build time ($T_{bs}$). This rate ($H_2$) could deviate from pre-processing, as environments during these operations vary, hence, the

overhead cost ($C_{OH_2}$) is calculated as:

$$C_{OH_2} = H_2 * T_{bs} \tag{11}$$

This ensures that overhead costs are accurately associated with the processing phase.

### c. Post-processing activity

Post-processing in AM entails finishing and inspection, including steps like heat treating. Post-processing costs cover material, machinery, labor, energy, inspection, and overhead. The formula for the post-processing cost is:

$$C_{pos} = C_{Mat_3} + C_{Mach_3} + C_{L_3} + C_{E_3} + C_T + C_{OH_3} \tag{12}$$

where:

$C_{pos}$ = post-processing cost ($)
$C_{Mat_3}$ = Material cost ($)
$C_{Mach_3}$ = Machine cost ($)
$C_{L_3}$ = Labor cost ($)
$C_{E_3}$ = Energy cost ($)
$C_T$ = Inspection cost ($)
$C_{OH_3}$ = Overhead cost ($)

This creates a full accounting and pricing of the costs necessary to undertake these final steps and produce a final product.

**Material cost.**  The first cost element of a post-processing activity is the material cost. You may need multiple materials depending upon customer requirements. This could include the costs of any chemicals or solutions you need to clean the parts or dissolve support.

Additionally, materials utilised for infiltrating, plating, or painting the parts must be considered. As such, this cost factor is defined as follows:

$$C_{Mat_3} = \sum_1^n W_i * CM_i \tag{13}$$

where:

$C_{Mat_3}$ = Cost of post-processing materials ($)
$W_i$ = Weight of the material (kg)
$CM_i$ = Cost rate of the material ($/kg) and $i$ = 1, 2, . . ., $n$.

This accounts for the variety and quantity of materials necessary to achieve the final product criteria.

**Machine cost.**  Beyond the 3D printing process itself, post-processing—another necessary part of most 3D printing production environments—can require a variety of different machines: each designed to perform an individual process, like curing or surface treatment. Machine expenses are tabulated according to the time a specific operation takes and the cost rate of the machine, which takes into account its initial purchase, expected lifespan, and salvage value. This approach ensures an accurate tabulation of machinery expenses related to post-processing:

$$C_{Match_3} = \sum_1^m M_j * T_j \tag{14}$$

where:

$$M_j = \text{Cost of utilizing machine } j \left(\frac{\$}{\text{h}}\right) = \frac{initial\ cost + exp - salvage}{Usful\ life}$$

$T_j$ = Time machine $j$ is used (h) $j$ = 1, 2, . . ., $m$.

**Labor cost.** The equation also captures the labor cost for post-processing activities; these can be labor-intensive, so the equation:

$$C_{L_3} = \sum_1^x T_q * O_q \tag{15}$$

where:

$T_q$ = Duration of work required (h)

$O_q$ = rate of operator cost (\$/h) $q$ = 1, 2, . . ., $x$.

ensures that the labor costs are accurately aligned with each operator's time and the knowledge and skill that that individual brings to the post-processing phase.

**Energy cost.** Cost for the post-processing machine's energy $C_{E_3}$, Cost of electricity used by machines such as Electrical Discharge Machines (EDM) or furnaces. This is calculated using:

$$C_{E_3} = \sum_{k=1}^y E_k * T_{bk} * E_{Rk} \tag{16}$$

Where:

$E_k$ = The machine uses $k$ energy, which is energy for M1 (Kw)

$T_{bk}$ = The $k$ time machine is in use (h)

$E_{Rk}$ = Cost of energy (\$/Kwh)

$y$ = The number of machines used in post-processing

This ensures energy costs accurately reflect the post-processing machinery's specific usage and energy rates.

**Testing cost.** Post-processing test costs, which may be considered for either 100% inspection or sampling inspection. The formula for sampling inspection is:

$$C_T = Q * A * n + (1 - Q)[(N - n) * A - p * N * B] \tag{17}$$

And for 100% inspection:

$$C_T = A * N + [p * N * B] \tag{18}$$

where:

$Q$ = Sample acceptance probability

$A$ = Cost per unit test (\$/unit)

$n$ = The number of sample units (unit)

$N$ = Total number of units

$p$ = Possible unit mismatch

$B$ = a single unit's maintenance or replacement cost (\$/unit).

These equations comprise of the expenses incurred in order to ensure that the quality of a part is good before it can be shipped. This includes all the inspections, audits and tests done to identify non-conforming goods

**Overhead cost.** The final part in calculating post-processing costs is overhead, denoted as which may differ from pre-processing rates because they might take place in different facilities

or locations. Overhead cost for post-processing:

$$C_{OH_3} = H_3 * T_{post} \tag{19}$$

where:

$H_3$ = cost rate of overhead ($/h)

$T_{post}$ = Duration of post processing (h)

This formula helps allocate accurately the general costs overhead specific to post processing phase.

## 3. Implementation

Companies can use the ABC model to evaluate AM costs for implant design. This model helps increase profitability while at the same time providing customers with a lower cost option for custom implants. The implementation approach involves collecting data, identifying cost drivers, calculating costs, allocating overhead, and determining total costs. Phalanges, which are small, flat, circular bones resembling the tips of fingers, serve various functions such as manipulating objects, grasping them, and tactile sensory perception. The distal phalanx, the last bone in a human finger, will serve as a case study for AM. In this study, we conducted an experiment at Sohar University's AM workshop in Oman, which houses multiple AM machines specialized in implant applications. Utilizing the MYSINT 100, a professional-grade machine, we fabricated a Distal Phalanx implant for a patient with a shattered bone, employing Stainless 316 as the material, shown in Fig 2.

This experiment aimed to validate predicted variables for the ABC model and explore the associated costs of metal printing using SLM technology. For accurate cost assessment of this process, it is crucial to first comprehend the machine's operation, detailed as follows:

1. Machine Preparation: The system is cleaned thoroughly, and the build platform is properly set.

2. Chamber Inertization: This step essentially involves the insertion of inert gas, for instance, argon or nitrogen inside the machine chamber such that the gas blanket efficiently excludes oxygen, a characteristic necessary for the purity of the melt and the prevention of oxidation.

3. Powder Loading: The printer's container is filled firstly with metal powder.

4. Powder Spreading: The metal powder is then evenly spread by re-coating in an ever-decreasing layer thickness over the build base, making it ready for thermoplastic fusion.

5. Laser Melting: The powder is selectively melted with the help of a laser beam, melting layer by layer until finally, it solidifies to form the geometry of the part.

6. Layer Deposition: In each fuse stage, the build platform is lowered, a new powder layer is applied, and the powder is fused together.

7. Continuous Inert Gas Flow: The gas flowing continuously along the line prevents the generation of an explosive atmosphere within the chamber.

8. Post-Build Cooling and Unfused Powder Removal: Once the building is over, the temperature is reduced and then followed by the removal of any leftover unfused powder, which is completed using a vacuum.

9. Powder Recycling: Finally, the sieving machine treats the unsintered powder, to remove the agglomerates and combine it with fresh powder to reuse it.

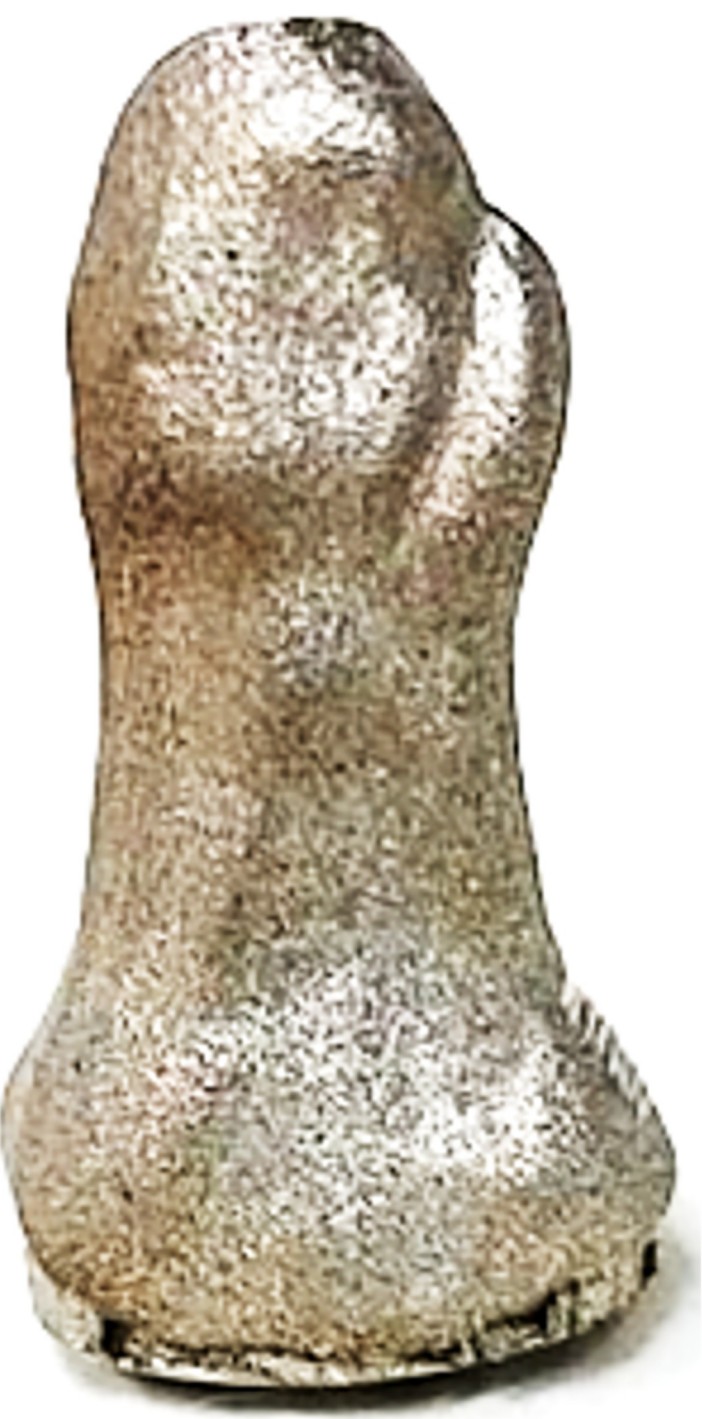

**Fig 2. The printed part.**

**Table 1. Pre-processing, processing, and post-processing cost estimation.**

| Shortened Form | Items | Value | Unit |
|:---:|:---|:---:|:---:|
| | **Preprocessing** | | |
| N | Numbers for parts | 1 | pcs |
| $O_1$ | Rate of operator cost | 14 | $/h |
| $T_{prep}$ | Duration of preparation | 2 | h |
| $H_1$ | Cost rate of overhead | 3.46 | $/h |
| | **Processing** | | |
| $W_1$ | Total actual powder consumed as build material | 0.112 | kg |
| $W_2$ | Inert gas consumption at 150 bar | 3.25 | liter |
| $CM_1$ | cost of materials build rate | 55.48 | $/kg |
| $CM_2$ | Cost of the inert gas at 150 bar | 4.08 | $/Liter |
| C1 | Value of Machine Purchase | 165,000 | $ |
| M | maintenance of machines | 27,000 | S |
| S | worth salvaging | 0 | $ |
| T | Practical Life Machine | 55,000 | h |
| $T_{bs}$ | Duration of Setup and Build | 2.067 | h |
| $O_2$ | Rate of Operator Cost | 10 | $/h |
| E | The AM System's Energy Use | 1.9 | kw |
| $T_b$ | Time to Build | 1.567 | h |
| $E_R$ | Cost of Energy Rate | 0.223 | $/kwh |
| $H_2$ | Rate of Production's Overhead Costs | 1.53 | $/h |
| | **Post-Processing** | | |
| Mj1 | Saw machine | 2.8 | $/h |
| Tj1 | It uses the time machine j (Sawing period). | 1 | min |
| $O_q$ | rate of operator cost | 8 | $/h |
| $T_q$ | Duration of work required | 0.25 | h/unit |
| $E_{k1}$ | The machine uses k energy, which is energy for M1. | 0.8 | Kw |
| $T_{bk}$ | The k time machine is in use | 1 | min |
| $E_{Rk}$ | Cost of energy | 0.223 | $/kwh. |
| A | Cost per unit test | 1.5 | $/unit |
| n | The number of sample units | 1 | Unite. |
| p | Possible unit mismatch | 1% | % |
| B | a single unit's maintenance or replacement cost | *70* | $/unit |
| $H_3$ | cost rate of overhead | 1.53 | $/h |
| $T_{post}$ | Duration of post processing | 0.25 | h |

Data summaries for each process are presented in Table 1. Let us now examine the cost breakdown of our case study and apply it to the ABC model:

## 3.1 Preprocessing activity

The preprocessing activity constitutes the initial step, encompassing the preparation of the CAD file until it is ready to be loaded into the machine and ensuring an error-free operation. In our case study, the production of the distal phalanx typically begins with acquiring CT scans of the damaged or shattered bone. These scans are then exported into DICOM format. Subsequently, the manufacturing team assumes the responsibility of converting these scans into a file prepared for the AM process. Specifically, the damaged distal phalanx scans are imported into Mimics software, which compiles the scans to generate a 3D model of the part.

This model is then exported as an STL file to 3-Matic and/or SolidWorks software to transform the 3D image into a 3D solid model, facilitating any necessary repairs to address defects and damages. Following this, the model, in STL format, is also exported to slicing software such as Netfabb and 3D Slicer. It is then loaded and configured using machine-specific software. In our case study, the preprocessing of the file, from receiving the DICOM file to loading the design into the machine, is conducted by a CAD engineer. The cost associated with this engineer is 14 USD per hour ($O_1$), with an average processing time of two hours ($T_{prep}$), as estimated by the workshop.

Regarding overhead costs, the preprocessing activity for AM incurs expenses related to software licenses for Mimics, 3-Matic, SolidWorks, and Netfabb, in addition to the depreciation and maintenance of computer hardware. Operational expenses, such as electricity and other related costs, are also considered. Furthermore, administrative overhead costs are included, encompassing expenses for building rental and maintenance, utilities, legal fees, and other related costs. These expenses are collectively calculated at 3.46 USD for the CAD department of the workshop at Sohar University. Eqs 2, 3 and 4 are used to calculate the preprocessing activity cost as:

$$C_{L_1} = O_1 * T_{prep} = 14 * 2 = \$28$$

$$C_{OH_1} = H_1 * T_{prep} = 3.46 * 2 = \$6.92$$

$$C_{Prep} = 28 + 6.92 = \$34.92$$

## 3.2 Processing activity

This begins after the processing activity and its main role is building the part and performing the AM manufacturing using the MYSINT 100 machine.

**a. Time to build and setup for the processing activity.** The calculated time to build the part $T_b$ is 1 hour and 34 minutes. This predicted time is calculated using Netfabb, Autodesk's toolset for AM used to estimate the build time, part's volume and then powder material cost, accordingly. The real time consumed in this case study is 1 hour and 32 minutes as per the machine generated reports of the process [S1 Appendix]. The Setup time before a build in SLM includes machine calibration, material and build chamber preparation, and loading job file. Depending on the workshop, these preparatory activities typically take around 30 minutes, ensuring the machine is ready for optimal operation and part production. Therefore, the setup and build time $T_{bs}$ is 2 hours and 4 minutes which is the total time the machine is powered on and will be used here to apply in equations of calculating processing cost. According to the report of the machine (see S1 Appendix) the printing of this part requires 1 hour and 59 minutes and 50 seconds.

**b. Materials.** The main materials used in the process are the metal powder and the inert gas. Stainless steel 316 powder is used to build the distal phalanx which is quoted by CARPENTER ADDITIVE at 55.48 USD/KG [17]. As stated above, the volume is calculated using Netfabb software which is 1.179 cm$^3$. The density of SS 316 is 7.99 g/cm$^3$. Therefore, the weight for the powder used to build this part is approximately 9.42 g. It is worth mentioning here that there is no technical need to add support material to this part therefore it is not used here in the implementation of the ABC model. However, this is not a correct calculation as outlined before regarding recycling the powder and mixing the recycled and fresh powder for another operation. In this case, 80% of the recycled powder is mixed with fresh powder to reuse for

another job and this is as recommended in the workshop where the experiment was conducted as per their experience. However, further studies are needed for each scenario and parameters to calculate the reuse possibility of the powder, such as those used in Gitajali and Mihaela's [18] study for the electron beam powder bed fusion with specific parameters all will thus help to deliver the best prediction calculation of the cost along with ensuring the quality of the built part.

In addition, as previously outlined with regards to the powder supply container and the build cylinder; we should go further to investigate how much (in terms of volume) powder the build cylinder of the machine accommodates, and this is dependent on two main factors in our case; the maximum height of the part and the build cylinder diameter. The height of the part is 20.66 mm which will be printed through 1033 layers, each one 0.02 mm. The build cylinder diameter using optional reduction of the build platform cylinder is 63.5mm in diameter and 34.5 mm height and it is worth mentioning that whenever there is an option for build platform reduction to minimum requirement of the part size it will be more cost effective to the operation, and as mentioned only 80% of the used powder could recycled. Only the build cylinder is considered here and not the supply cylinder because after the job is completed, the powder that will be subjected to the reused percentage as stated above is only the powder that remains in the build cylinder which undergoes fusion but the remaining in the supply cylinder chamber is not subjected to this percent and is considered 100% reusable. Therefore, in this case the material cost will be calculated as below.

$$V_t = -V_{part} + \left[ \left( B_{Ch} - V_{part} \right) x \left( 1 - R_{percent} \right) \right]$$

Where:

$V_t$ is the volume of the powder used to produce this part which includes the part itself and the affected percent of the remaining powder which are all considered bearing the cost of the produced part.

- $V_{part}$ is the volume of the part which is 1.179 cm$^3$

- $B_{ch}$ is the volume of the build chamber with max high that is reached to print this part which is here 65.455 cm$^3$ (where the diameter of the cylinder is 6.35 cm and height of the build is 2.066 cm)

- $R_{percent}$ reuse possibility is 80%

  Therefore:

$$Vt = 1.179 + [(65.455 - 1.179) \text{ x } (1 - 0.8)]$$

$Vt$ = 14.034, and therefore $W_1$ weight of the build material that is used as per the ABC model is 112.13 g and here as total actual powder consumed as build material. The second material is the inert gas which in SLM machines like the SISMA MYSINT100 is used to prevent oxidation during the melting process, thus ensuring high-quality prints. Initially, the chamber is filled with inert gas to displace any oxygen-rich air (inertization), creating an inert atmosphere. Throughout the build, a continuous flow of gas maintains this environment, preventing contamination and facilitating optimal laser-material interaction. With this machine and the chosen powder material (ss 316), the inert gas that should be used is Nitrogen with a purity of 99.999 as recommended for this medical case application.

As stated above, the consumption of the inert gas is in two stages; firstly, before the beginning of the building stage, the chamber should be filled with the inert gas which is 410 liter for this specific machine and the second stage is during the building time which is calculated to 50

liters per hour of build time. While in this case study, the real consumption was 407.92 liters for machine chamber inertization and 80.4 liters for 1 hour and 32 minutes of build time. In this case, a nitrogen gas cylinder with purity of 99.999 of 40 liters at 150 bar manufactured by BrotherGas is quoted [19] at 163.2 USD. For estimated average consumption as per manufacturer for chamber inertization at 410 liters and with built time predicted by Netfabb at 1 hour and 34 minutes (where 50 liter per hour will be consumed) the total required liters at 1 bar is 488.33 liters and when applying Boyle's law it is calculated that the cylinder will consume 3.25 liter at 150 bar ($W_2$). Despite that Eq 6 is used when there is multiple material used for build and support but there is no support material in this case, the same equation is still applicable, where:

$$C_{MAT} = (W_1 * CM_1) + (W_2 * CM_2) = (0.112 * 55.48) + (3.25 * 4.08) = \$19.47$$

**c. Machine cost.**　The value of the machine purchase ($C_1$) is \$165,000 as quoted by the manufacturer (SISMA) according to the same conditions that experiment was conducted in, while the lifespan of the machine in terms of working hours is 70,000 hours as claimed by the manufacturer however this lifespan is considered with highest level of maintenance practice, types of materials and adhering to the manufacturer recommendation therefore in this study a lifespan of 55,000 hours ($T$) is considered to reflect the realistic user commitment to the manufacturer recommendations of maintenance and usability.

While the calculated maintenance cost as per the manufacturer for this lifespan is \$27000 (M) which encompassing Preventive Maintenance to check and replace wear parts like filters, recoated blades, and protective window lenses. Preventive maintenance might also include calibration and software updates and encompassing consumable's maintenance which includes replacement of filters, gas nozzles, and other components that have a limited lifespan. In addition to unexpected repairs, despite regular maintenance, unexpected issues may arise that require repairs.

In this study, the salvaging worth (S) is set to zero due to technological obsolescence which often outpaces physical wear, significantly diminishing the resale value of specialized machinery like SLM printers. A limited market for such high technology used equipment and the potential costs associated with disposal or refurbishment further supports this assumption, ensuring a conservative approach in the cost modeling process using Eq 8, as follow:

$$C_{Mach} = \left(\frac{C_1 + M - S}{T}\right) \times T_{bs} = \left(\frac{165,000 + 27,000 - 0}{55,000}\right) \times 2.067 = \$7.215$$

**d. Labor cost.**　The operator of the machine and the related activity to build the part is estimated by the workshop as \$10 ($O_2$), using Eq 9 will be:

$$C_{L_2} = O_2 * T_{bs} = 10 * 2.067 = \$20.67$$

**e. Energy.**　According to the technical details for the SISMA MYSINT 100 [20], the maximum power level during a build is 1.9 kw which is considered the worst-case scenario for this printer. The cost of the energy rate will be considered as 0.223 \$/kwh which is the energy rate for businesses in Oman [21], using Eq 10.

$$C_E = E * T_b * E_R = = 1.9 * 2.067 * 0.223 = \$0.875$$

**f. Overhead cost.** The overhead cost in the processing activity will be considered mostly here as the administrative cost as mentioned before at 1.53 \$/h. as per Eq 11:

$$C_{OH_2} = H_2 * T_{bs} = 1.53 * 2.067 = \$3.162$$

Therefore, the processing cost as per Eq 5 will be:

$$C_{processing} = C_{MAT} + C_{Mach} + C_{L_2} + C_E + C_{OH_2} = 19.47 + 7.215 + 20.67 + 0.875 + 3.162$$
$$= 51.392$$

## 3.3 Post processing activity

The post processing activity in SLM with this machine may encompass many operations such as cutting to cut the part from the support materials and/or the build platform and surface finishing operation and heat treatment in addition to quality control and testing activity, however due to shape and simplicity of this part there is no advanced operations subjected to the printed parts. In this case, a bandsaw machine is used to cut the part and a surface finish with hand tools is made only. The other data collected from the workshop is presented in Table 1 and the estimated cost will be shown in Eqs 12, 14, 15, 16, 18 and 19:

$$C_{Match_3} = \sum_1^m M_j * T_j == 2.8 * 0.016 = \$0.046$$

$$C_{L_3} = \sum_1^x T_q * O_q = 0.25 * 8 = 2$$

$$C_{E_3} = \sum_1^y E_k * T_{bk} * E_{Rk} = 0.8 * 0.016 * 0.223 = \$ 0.0028$$

$$C_T = A * N + [p * N * B] = 1.5 * 1 + [0.01 * 1 * 70] == 1.5 * 1 + [0.01 * 1 * 100] = \$2.2$$

$$C_{OH_3} = H_3 * T_{post} = 1.53 * 0.25 = \$0.382$$

$$C_{pos} = C_{Mach_3} + C_{L_3} + C_{E_3} + C_T + C_{OH_3} = 0.046 + 2 + 0.003 + 2.2 + 0.382 = \$4.631$$

Therefore, the total cost will be determined using Eq 1:

$$Total_{Cost} = \frac{1}{N} * \left( C_{prep} + C_{processing} + C_{pos} \right) = \left( \frac{1}{1} \right) * (34.92 + 51.392 + 4.631) = \$90.943$$

Fig 3 shows that creating activity cost breakdown entails providing a full account of the cost components associated with each activity and evaluating their impact on the total cost. This analysis is done by entering the relevant information into a spreadsheet containing columns for activity, cost, and percentage of total cost. The Percentage column represents the relative impact of each activity on the total cost. It is worth noting that the processing cost has the largest impact on the total cost, representing 57%, followed by the pre-processing cost (38%), and lastly the post-processing cost (5%) shown in Table 2.

This breakdown provides a complete overview of how costs are distributed among different activities and how they affect the overall budget. To graphically display this cost distribution, a

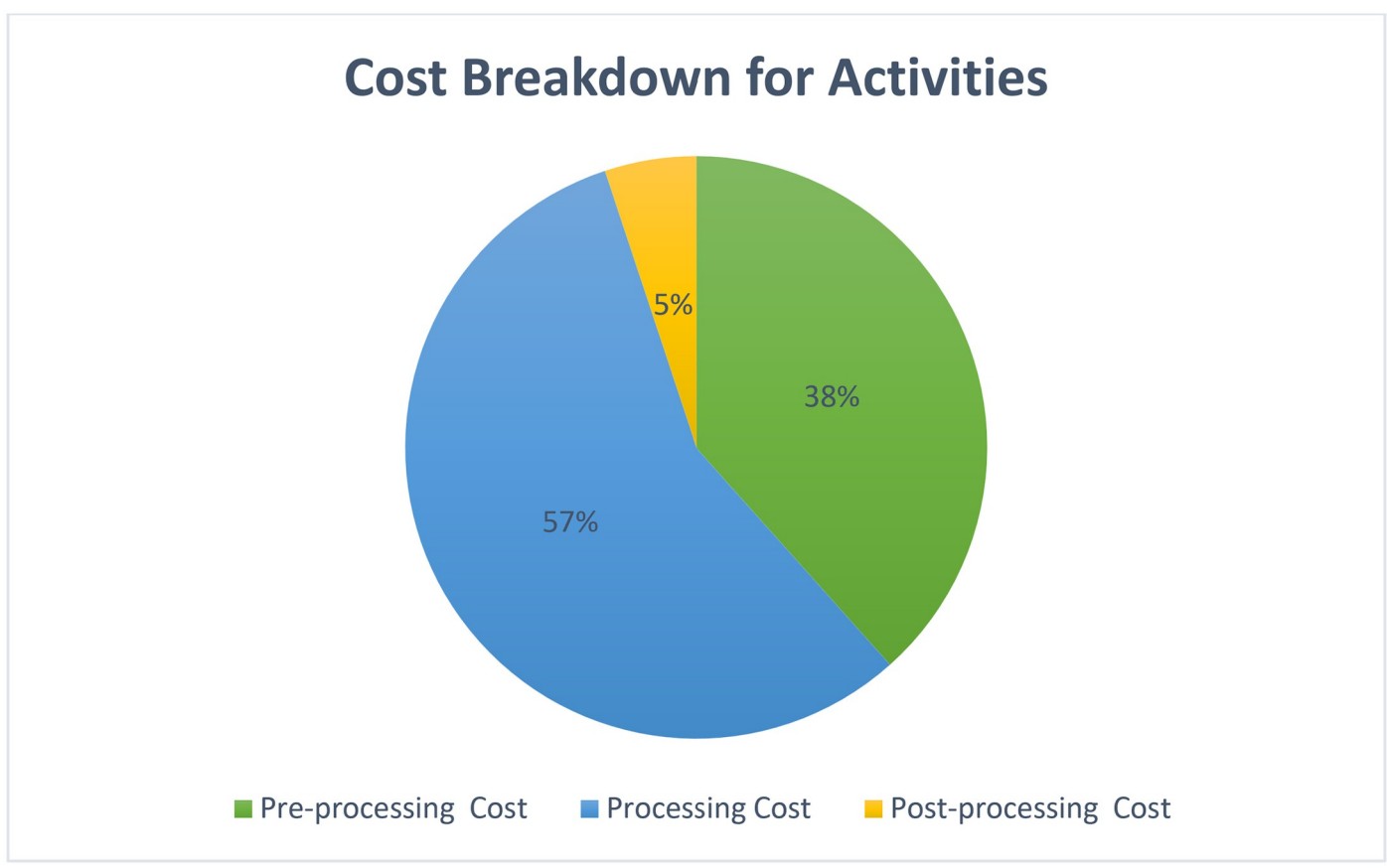

**Fig 3. Cost breakdown for activities.**

pie chart was created to show the percentage distribution for each activity within the total cost. The resulting pie chart represents the exact cost breakdown, with each activity represented by a segment according to its contribution to the total cost. This graphical representation provides a quick and clear depiction of how expenses are allocated across different activities.

Fig 4 shows the cost chart by category across all activities including labor, machines materials, overhead, energy and other factors that most impact the total cost illustrated in Table 3. This breakdown assumes that cost components are distributed across these categories. The percentages represent the ratio of each category's cost to the total cost, and the Impact on Total Cost column indicates the extent to which each category affects the total cost. The result was the perception that labor is the most influential category, followed by materials, overheads, machinery, etc., and energy in the order of highest influence.

**Table 2. Cost breakdown for activities.**

| Activity | Percent | Amount |
|---|---:|---|
| Pre-processing Cost | 38% | 34.92 |
| Processing Cost | 57% | 51.392 |
| Post-processing Cost | 5% | 4.631 |
| **Total Cost** | | **90.943** |

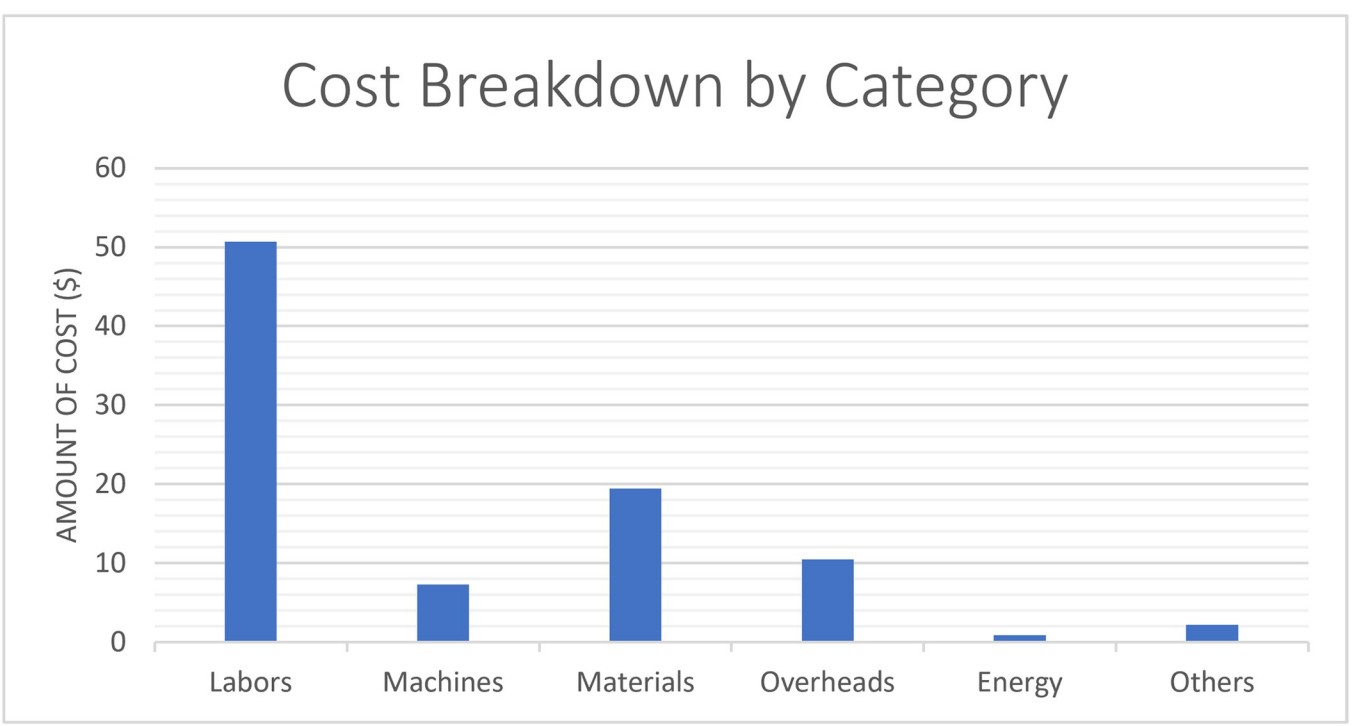

**Fig 4. Cost breakdown by category.**

Fig 5 shows the chart of Cost Breakdown by Highest Elements that include the Labor–Preprocessing, Overhead—Preprocessing, Powder, Nitrogen, Machine—process, Labor—Process, Overhead—process and Other. This breakdown assumes that the cost components are distributed across these specific elements. The percentages represent the proportion of each element's cost to the total cost, and the "Impact on Total Cost" column indicates the extent of each element's influence on the overall cost illustrated in Table 4. The results show that the three highest influential elements include the Labor—Preprocessing (the Highest Element 31%) followed by the labor processing (23%) and the Nitrogen influential (15%).

The two charts show the cost breakdown by category and the cost breakdown by Highest Elements and determine which of the parameters are the highest and most influential, while leaving the rest ineffective and called others.

Finally, A financial model, known as sensitivity analysis, is used to ascertain the effect of input variables on target variables. This analysis serves as a means of predicting decision outcomes, considering a specific set of variables. Table 5 shows the important variables selected as

**Table 3. Cost breakdown by category.**

| Category | $ | % |
|---|---|---|
| Labors | 50.67 | 56% |
| Machines | 7.261 | 8% |
| Materials | 19.47 | 21% |
| Overheads | 10.464 | 12% |
| Energy | 0.88 | 1% |
| Others | 2.195 | 2% |

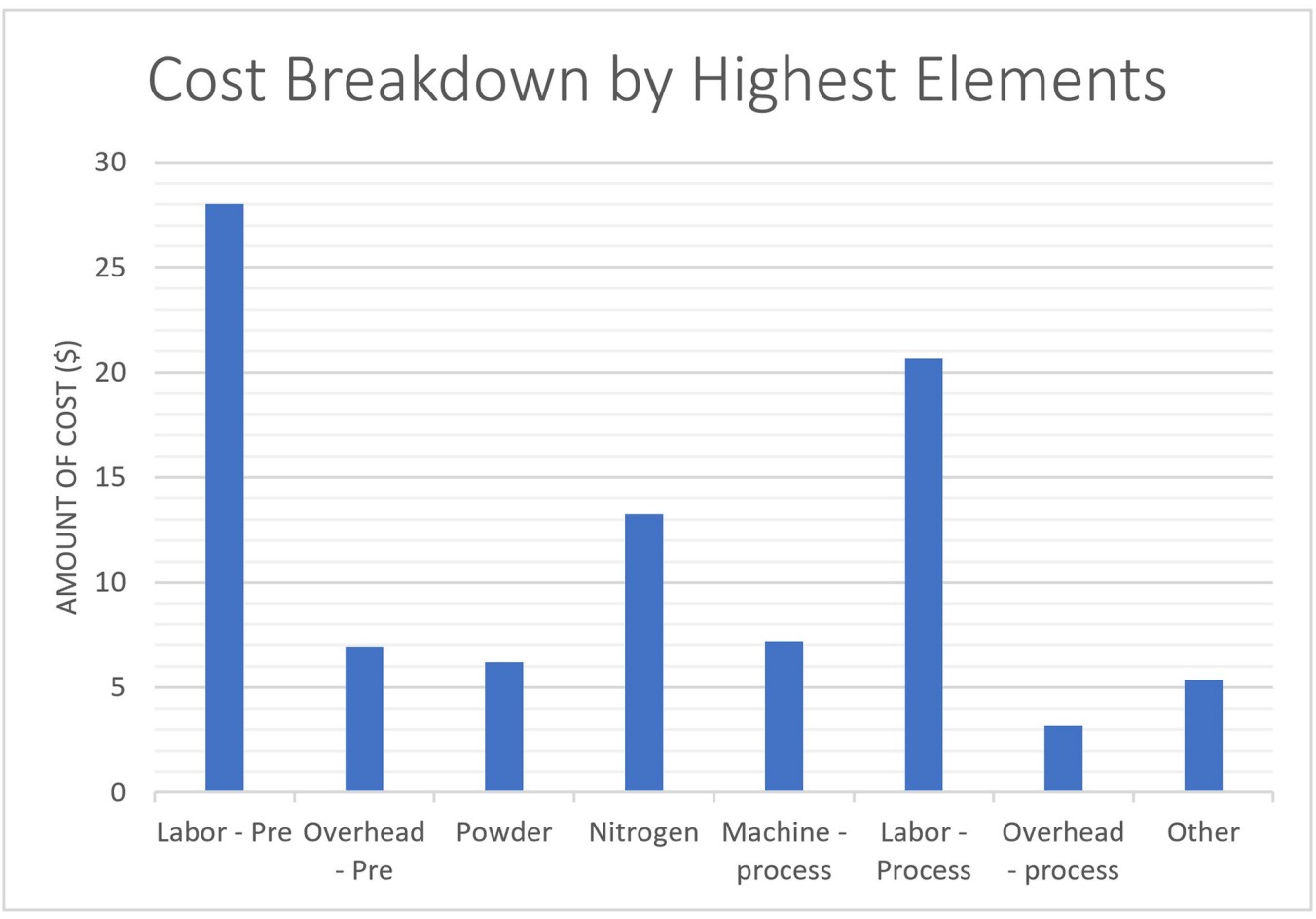

**Fig 5. Cost breakdown by highest elements.**

each influencing variable has been changed in certain proportions, and it can be seen how much it affects the total cost calculated to determine the risks it faces in the future. The percentages were changed from 10% to 40% or reduced from -10% to -40%. With any change in the percentage, the extent of its impact on the total cost in percentage and number is seen. In this study, a sensitivity analysis was performed to clarify the influences that could affect the final cost and the result shows that $O_1$ had a significant impact. When the increase is 40%, the

**Table 4. Cost breakdown by highest elements.**

| Highest Elements | $ | % |
|---|---|---|
| Labor - Pre | 28 | 31% |
| Overhead - Pre | 6.92 | 8% |
| Powder | 6.21 | 7% |
| Nitrogen | 13.26 | 15% |
| Machine - process | 7.21 | 8% |
| Labor - Process | 20.67 | 23% |
| Overhead - process | 3.162 | 3% |
| Other | 5.378 | 6% |

**Table 5. The result of sensitivity analysis.**

| | Base Case | 10% | 20% | 30% | 40% |
|---|---|---|---|---|---|
| **Rate of operator cost—Pre ($O_1$) ($/h)** | 14 | 15.4 | 16.8 | 18.2 | 19.6 |
| **Total Cost ($)** | $90.95 | $93.75 | $99.91 | $111.00 | $130.22 |
| | | 3.1% | 9.9% | 22.0% | 43.2% |
| | Base Case | 10% | 20% | 30% | 40% |
| **Rate of operator cost—Process ($O_2$) ($/h)** | 10 | 11 | 12 | 13 | 14 |
| **Total Cost ($)** | $90.95 | $93.02 | $97.56 | $105.75 | $119.94 |
| | | 2.3% | 7.3% | 16.3% | 31.9% |
| | Base Case | 10% | 20% | 30% | 40% |
| **Rate of operator cost—Post ($O_q$) ($/h)** | 8 | 8.8 | 9.6 | 10.4 | 11.2 |
| **Total Cost ($)** | $90.95 | $91.15 | $91.59 | $92.38 | $93.75 |
| | | 0.2% | 0.7% | 1.6% | 3.1% |
| | Base Case | 10% | 20% | 30% | 40% |
| **Cost of Build Material Rate ($CM_1$) ($/kg)** | 55.48 | 61.028 | 66.576 | 72.124 | 77.672 |
| **Total Cost ($)** | $90.95 | $91.57 | $92.94 | $95.40 | $99.66 |
| | | 0.7% | 2.2% | 4.9% | 9.6% |
| | Base Case | 10% | 20% | 30% | 40% |
| **Cost of the Inert Gas at 150 bar ($CM_2$) ($/Liter)** | 4.08 | 4.488 | 4.896 | 5.304 | 5.712 |
| **Total Cost ($)** | $90.95 | $92.28 | $95.19 | $100.44 | $109.55 |
| | | 1.5% | 4.7% | 10.4% | 20.4% |
| | Base Case | 10% | 20% | 30% | 40% |
| **Maintenance of Machine (M) ($)** | $27,000 | $29,700 | $32,400 | $35,100 | $37,800 |
| **Total Cost ($)** | $90.95 | $91.05 | $91.27 | $91.68 | $92.37 |
| | | 0.1% | 0.4% | 0.8% | 1.6% |
| | Base Case | 10% | 20% | 30% | 40% |
| **Cost of Energy Rate—Process (ER) ($/kwh)** | 0.223 | $0.25 | $0.27 | $0.29 | $0.31 |
| **Total Cost ($)** | $90.95 | $91.04 | $91.23 | $91.58 | $92.18 |
| | | 0.1% | 0.3% | 0.7% | 1.4% |
| | Base Case | -10% | -20% | -30% | -40% |
| **Inert Gas Consumption at 150 bar ($W_2$) (Liter)** | 3.25 | 2.925 | 2.6 | 2.275 | 1.95 |
| **Total Cost ($)** | $90.95 | $89.62 | $87.24 | $84.37 | $81.70 |
| | | -1% | -4% | -7% | -10% |

difference is 43.2%. $O_2$ also has a strong effect when increased by 40%, causing a difference of 31.9%. Also, it is clear that the effect of $O_q$ is small because it is rarely used. Machine maintenance was not affected by the 40% increase, as the difference was 1.6%, W2 is the inert gas consumed during the process. The percentage was calculated in the case of a negative decrease to prove that recycling costs in the future will be very low, as when the decrease is -40%, the loss or cost reduction is only -10%, but in the case A -90% reduction will not lose anything with regards to cost, so the goal is to see which changes are most sensitive or risky to the overall cost.

## 4. Conclusions

A simplified action-based costing model for medical Custom Implants produced through AM technology is developed in this study. The developed model is validated using a case study to estimate the manufacturing cost of a human finger's phalanges bone. Results has shown to be acceptable and encourage to use such model in wider applications. The model requires to

identify the components of the pre-processing, processing, and post-processing operations with estimation of the required time and cost of each component of the whole process. In the conducted case study for the manufacturing of human finger distal phalanx implant, the cost and time were calculated using the developed model. Hence the developed ABC model has successfully calculated the total cost. Results has shown that manufacturers should allocate $90.95 for the AM build process for the metal model of the part considered in this study. Part processing costs represented 57%, Pre-processing cost represented 38%, and post-processing cost represented only 5% of the total cost with 8%, 12%, 1% attributed to machine overhead and energy. Material costs accounted for only 21%, while labor costs accounted for the highest percentage, 56%. These results demonstrate their validity for the studied case, therefore, the use of the developed model in wider applications is recommended, though attention should always be given to the details of the cost components that suit the application to enable the model provides more reliable cost estimates and better business decisions.

## Supporting information

**S1 Appendix.**
(PDF)

## Author Contributions

**Conceptualization:** Zaineb Hameed Neamah.

**Data curation:** Zaineb Hameed Neamah.

**Formal analysis:** Zaineb Hameed Neamah.

**Methodology:** Zaineb Hameed Neamah.

**Resources:** Zaineb Hameed Neamah.

**Software:** Zaineb Hameed Neamah.

**Supervision:** Luma A. H. Al-Kindi, Ghassan Al-Kindi.

**Writing – original draft:** Zaineb Hameed Neamah.

**Writing – review & editing:** Luma A. H. Al-Kindi, Ghassan Al-Kindi.

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
