## [Decision Letter · Decision Letter 0]

9 Jan 2024

PONE-D-23-42855ABC Model for Cost Estimation of Custom Implants by Additive ManufacturingPLOS ONE

Dear Dr. hameed,

Thank you for submitting your manuscript to PLOS ONE. After careful consideration, we feel that it has merit but does not fully meet PLOS ONE’s publication criteria as it currently stands. Therefore, we invite you to submit a revised version of the manuscript that addresses the points raised during the review process.

We look forward to receiving your revised manuscript.

Kind regards,

Saliha Karadayi-Usta, PhD

Academic Editor

PLOS ONE

Journal Requirements:

Additional Editor Comments (if provided):

Please take the below reviewer suggestions into consideration and make necessary revisions.

Reviewers' comments:

Reviewer's Responses to Questions

**Comments to the Author**

1. Is the manuscript technically sound, and do the data support the conclusions?

Reviewer #1: Partly

Reviewer #2: Yes

Reviewer #3: Yes

Reviewer #4: Partly

2. Has the statistical analysis been performed appropriately and rigorously? 

Reviewer #1: Yes

Reviewer #2: Yes

Reviewer #3: Yes

Reviewer #4: No

3. Have the authors made all data underlying the findings in their manuscript fully available?

Reviewer #1: Yes

Reviewer #2: Yes

Reviewer #3: Yes

Reviewer #4: Yes

4. Is the manuscript presented in an intelligible fashion and written in standard English?

Reviewer #1: Yes

Reviewer #2: No

Reviewer #3: Yes

Reviewer #4: No

5. Review Comments to the Author

Reviewer #1: As a reviewer, I have carefully examined the ABC Model for Cost Estimation of Custom Implants by Additive Manufacturing. My major comments and observations are outlined below:

1. Provide additional details on the mathematical formulation of the ABC model. A more explicit presentation of equations, variables, and their relationships would enhance the technical understanding of the model.

2. Specify the sources of data used for parameter estimation in the cost model. Additionally, consider discussing any validation processes employed to ensure the accuracy and reliability of the input data.

3. Further elaborate on the breakdown of costs within each phase (pre-processing, processing, post-processing). For instance, elaborate on how overhead costs are calculated and allocated in each step.

4. Consider conducting a sensitivity analysis to evaluate the impact of variations in key parameters on the overall cost estimates. This could include variations in material costs, machine maintenance, or energy costs.

5. Clearly state any assumptions made during the case study, such as machine lifespans, maintenance costs, and energy consumption. This will enhance the transparency of the study and allow readers to assess the generalizability of the findings.

6. Provide metrics for comparing the accuracy of the proposed ABC model with alternative cost estimation approaches. This could include a comparison with traditional manufacturing cost models to highlight the advantages of the proposed method.

7. Discuss how external factors, such as market fluctuations or regulatory changes, might influence the accuracy of the cost estimates. Understanding the model's sensitivity to external variables is crucial for real-world applicability.

8. The time estimates for various phases of the additive manufacturing process could be further detailed. Include considerations for potential variations in production times based on factors like technology advancements or process optimizations.

9. Discuss the rationale behind the selection of specific materials (plastic and metal) for the case study. Consider including a discussion on the potential variations in cost estimates with different materials.

10. Consider incorporating visual aids such as charts or graphs to represent the cost breakdown more intuitively. Visualizations can enhance the reader's understanding and facilitate a quicker grasp of complex information.

11. Provide more information about the specific additive manufacturing machines used in the case studies, including their capabilities, limitations, and any unique features that might impact costs.

12. Discuss how the proposed cost model can be integrated into existing manufacturing workflows. Considerations for user-friendliness and adaptability to different AM systems would be valuable information.

Reviewer #2: Reviewer’s comments

The content of the paper is practical and has more use to many stake-holders such as patients, manufacturers, doctors and salespersons.

Thus, calculations should be more clear and add table of comparison allowing for better and easy selection of items.

1. Grammatical mistakes are to be corrected

a. Yang and Li had [5] developed – should be in present simple tense

b. Ulu et al. (2019) [6] suggested - should be in present simple tense

2. Referencing should be corrected, in some places only the [number] but in some cases both names and number as in the example.

a. Yang and Li had [5]

b. Ulu et al. (2019) [6]

3. The total cost (PT) should be corrected as 

4. The term used in research for ‘Shortened form’ is Abbreviation.

5. Move “Table 1: Pre-processing, Processing, and Post-processing Cost Estimation” before the calculations as placed as the Table 2.

6. PETG has not been defined.

7. Formatting errors were notified in few places.

8. Mistakes in the List of References has to be corrected;

a. Name of the journal missing: [16], [17]

b. et al.

9. Start from a Capital letter: [2]

Reviewer #3: This work shows basic work of cost model for AM processes. The work shows an economical model developed with two case studies. The model is simple and it could be applied to different AM processes. AM processes are not as difficult as other manufacturing this is why the cost model is simple.

I have one consideration to be taken in account.

1. Use Nowadays instead of Now a days

Reviewer #4: * The keywords should be improved, considering that the authors are actually proposing ABC method. Furthermore, what does FDM stand for?

* Usually, the acronyms defined in the abstract, should be re-defined in the manuscript (outside of the abstract)

* The way the abstract is organized suggest that the authors are suggesting ABC method, however, based on the introduction, it seems that there are existing ABC models suggested in the literature. Which brings these two questions: What are the novelties of this study? Is this paper proposing the ABC model for the first time?

* The written quality of sections 2 and 3 are not acceptable and must be improved.

* What does m refer to in equation 1?

* What are the contributions of this study? In another word, how can another study implement the proposed model of this study, given the model is generic. Furthermore, the case study is so specific which makes it particularly hard to implement the result of this study in other studies.

* There are no descriptions provided for the mathematical models (what are the parameters in the equations?)

* What are the limitations of this study?

6. PLOS authors have the option to publish the peer review history of their article (what does this mean?). If published, this will include your full peer review and any attached files.

Reviewer #1: No

Reviewer #2: No

Reviewer #3: No

Reviewer #4: No

---

## [Author Response · Author response to Decision Letter 0]

4 Mar 2024

Title: ABC Model for Cost Estimation of Custom Implants by Additive Manufacturing

Manuscript Number: PONE-D-23-42855

Dear Respected Reviewers,

First of all, we would like to express our sincere gratitude and appreciation to the valuable feedback provided by the Reviewers. Secondly, attached please find the revised manuscript “ABC Model for Cost Estimation of Custom Implants by Additive Manufacturing”. Reviewers’ comments were addressed in this version. We hope that it will meet the editors’/reviewers’ expectation and be accepted for publication in PLOS ONE Journal.

We have carefully considered each comment provided by the reviewers and most the provided comments and suggestions were implemented in the revised manuscript. We believe that these comments and suggestions have improved the overall quality of the paper. 

In the next few pages, we have provided a detailed response to each comment/suggestion provided by the reviewers. For clarity of exposition, all changes in the manuscript are highlighted in blue.

Yours sincerely,

Authors

First of all, we would like to thank the reviewers for their valuable comments and suggestions. Please find our detailed responses in blue.

Reviews **********************

Reviewer 1:

As a reviewer, I have carefully examined the ABC Model for Cost Estimation of Custom Implants by Additive Manufacturing. My major comments and observations are outlined below:

1. Provide additional details on the mathematical formulation of the ABC model. A more explicit presentation of equations, variables, and their relationships would enhance the technical understanding of the model.

Authors response: Thanks to the reviewer for his valuable comment. Following the reviewer's recommendation, we have enhanced the technical understanding of the ABC by providing more details on its mathematical formulation. The equations are clearly discussed, and the variables are adequately defined. Additionally, the relationship between the equations are provided. The details mathematical formulation of the ABC algorithm is illustrated in Section Two (‘The Methodology of Total Cost Model Calculations’, pages 2-9 of the revised manuscript).

2. Specify the sources of data used for parameter estimation in the cost model. Additionally, consider discussing any validation processes employed to ensure the accuracy and reliability of the input data.

Authors response: Thanks to the Reviewer for his valuable comments. We would like to mention that all data were taken from the workshop at Sohar University (please refer to Appendix 1).

3. Further elaborate on the breakdown of costs within each phase (pre-processing, processing, post-processing). For instance, elaborate on how overhead costs are calculated and allocated in each step.

Authors response: Thanks to the Reviewer for his important comments, which alerted us that the cost calculations were not adequately addressed. Thus, following the reviewer's recommendation, this issue is further elaborated in the revised manuscript. Specifically, in Section Two (The Methodology of Total Cost Model Calculations), additional discussion has been included to break down the cost calculation for the three main activities (pre-processing, processing, post-processing) and their associated cost components shown in Figure 1 (pages 2-9 of the revised manuscript). 

4. Consider conducting a sensitivity analysis to evaluate the impact of variations in key parameters on the overall cost estimates. This could include variations in material costs, machine maintenance, or energy costs.

Authors response: Thanks to the Reviewer for his insightful suggestion. Sensitivity analysis was conducted a to assess the impact of variations in key parameters on our overall cost estimates. It has been clarified how changes in critical factors may influence the reliability and robustness of our cost model, as showed in Table 1 (please refer to page 19 of the revised manuscript).

5. Clearly state any assumptions made during the case study, such as machine lifespans, maintenance costs, and energy consumption. This will enhance the transparency of the study and allow readers to assess the generalizability of the findings.

Authors response: We appreciate the reviewer's insightful suggestion to explicitly state the assumptions made during our case study. All data were factual, but Assumptions regarding routine maintenance costs were made using industry benchmarks and expert consultations by the workshop at Sohar University. We only paid $55 to maintain the machine, while the machine manufacturer charged $70 because machine hours were limited to standard conditions. it's important to note that actual maintenance costs can vary based on specific operational practices and the condition of individual machines Therefore, considering the conditions were not standard, readers should assess the applicability of our findings in the context of their own maintenance protocols (please refer to Table 1 on page 11).

6. Provide metrics for comparing the accuracy of the proposed ABC model with alternative cost estimation approaches. This could include a comparison with traditional manufacturing cost models to highlight the advantages of the proposed method.

Authors response: We appreciate the suggestion to provide metrics for comparing the accuracy of our proposed Activity-Based Costing (ABC) model with alternative cost estimation approaches, including traditional manufacturing cost models. Thank you for your valuable feedback, and we look forward to incorporate these enhancements into future work.

7. Discuss how external factors, such as market fluctuations or regulatory changes, might influence the accuracy of the cost estimates. Understanding the model's sensitivity to external variables is crucial for real-world applicability.

Authors response: To address the Reviewer’s concern, the cost estimation model is adjustable and reactive to external factors. It can modify cost estimates based on market dynamics and regulatory changes. Sensitivity analysis increases the model's real-world applicability by considering the dynamic nature of the business environment. The machine can be used without any restrictions in appropriate laboratories with all necessary safety procedures.

8. The time estimates for various phases of the additive manufacturing process could be further detailed. Include considerations for potential variations in production times based on factors like technology advancements or process optimizations.

Authors response: We appreciate the reviewer's feedback and acknowledge the importance of providing detailed time estimates for each phase of the additive manufacturing process. In our revised work, we enhanced the time estimates and explained it in detail (please refer to Section 3), where we relied on the program used for simulating and on the report of the manufacturing process and machine. This will provide a more nuanced understanding of the dynamic nature of additive manufacturing timelines.

9. Discuss the rationale behind the selection of specific materials (plastic and metal) for the case study. Consider including a discussion on the potential variations in cost estimates with different materials.

Authors response: In selecting plastic and metal materials for the case study, we aimed to represent a common and diverse set of materials used in additive manufacturing. In our revised work, we wanted to make the research more realistic. However, plastic would no longer be useful for real-world application but would serve merely as an experimental part. Therefore, we limited ourselves to use only metal, as it is medically and practically suitable for the patient.

10. Consider incorporating visual aids such as charts or graphs to represent the cost breakdown more intuitively. Visualizations can enhance the reader's understanding and facilitate a quicker grasp of complex information.

Authors response: To address the Reviewer’s concern, we appreciate the suggestion to enhance clarity. In response, we have incorporated visual aids with three charts to represent the cost breakdown more intuitively. The first one is the pie chart. As for the remaining two, we analyzed them and clarified the cost picture, showing where it is concentrated, its importance, and what affects the equation in general, as illustrated in Figures 3-5 on pages 16-18 of the revised manuscript. This addition aims to improve the reader's understanding and facilitate a quicker grasp of complex information in our revised presentation.

11. Provide more information about the specific additive manufacturing machines used in the case studies, including their capabilities, limitations, and any unique features that might impact costs.

Authors response: Thank you for the valuable feedback. In our case study, we are utilizing the MYSINT 100 for manufacturing custom implants, offering advantages such as high precision, flexibility in design, and efficient production. Its selective laser melting technology allows for intricate and patient-specific implant designs, contributing to better fit and enhanced functionality. The MYSINT 100's capabilities in processing various materials make it suitable for a range of implant applications, providing a reliable and customizable solution in additive manufacturing for medical purposes.

12. Discuss how the proposed cost model can be integrated into existing manufacturing workflows. Considerations for user-friendliness and adaptability to different AM systems would be valuable information.

Authors response: We appreciate the reviewer's input. In addressing this, indeed, the proposed cost model can seamlessly integrate into existing manufacturing workflows. The factory owner can achieve this by creating an Excel sheet template containing all the available information on materials and gas etc., and then implementing the cost model. This approach considers the practical implementation of the model.

Reviewer 2:

The content of the paper is practical and has more use to many stake-holders such as patients, manufacturers, doctors and salespersons. Thus, calculations should be more clear and add table of comparison allowing for better and easy selection of items.

Authors response: Thanks to the Reviewer for his important comments, which alerted us that the cost calculations were not adequately addressed. Thus, in the revised manuscript, this issue is further elaborated. Specifically, additional discussion has been included to break down the cost calculation for each activity, depending on its main actions (pages 2-9 of the revised manuscript).

1. Grammatical mistakes are to be corrected

 Yang and Li had [5] developed – should be in present simple tense

 Ulu et al. (2019) [6] suggested - should be in present simple tense

Authors response: To address the Reviewer’s concern, all minor edits have been completed, and the manuscript was proofread by a native English speaker from Canada. Consequently, all the typographical and grammatical errors have been essentially eliminated. Also, please refer to page (2) of the revised manuscript for the mistakes mentioned above.

2. Referencing should be corrected, in some places only the [number] but in some cases both names and number as in the example.

a. Yang and Li had [5]

b. Ulu et al. (2019) [6]

Authors response: Thanks to the Reviewer for his valid comment. The authors have unified the way of referring to the references by using only [Number](please refer to page 2 of the revised manuscript).

3. The total cost (PT) should be corrected as 𝑃𝑇

Authors response: Many thanks to the Reviewer for this comment. The total cost has been denoted by 〖Total 〗_Cost in the revised manuscript (please refer to page 3). 

4. The term used in research for ‘Shortened form’ is Abbreviation.

Authors response: Thanks to the Reviewer for his valid comment. The authors have changed the term in Table 1 from ‘Shortened form’ to ‘Abbreviation’ in the revised manuscript (please refer to page 11).

5. Move “Table 1: Pre-processing, Processing, and Post-processing Cost Estimation” before the calculations as placed as the Table 2.

Authors response: Thanks to the Reviewer for his valid comment. Table 1 is no longer used in the revised manuscript, as explained in the following response. 

6. PETG has not been defined.

Authors response: Thanks to the reviewer for his valid comment. PETG stands for PolyEthylene Terephthalate Glycol, a commonly used additive manufacturing material. In our revised work, we aimed to make the research more realistic, and plastic would no longer be suitable for real application but merely for experimental parts. For this reason, we limited ourselves to using metal only because it is medically and practically suitable for patients.

7. Formatting errors were notified in few places.

Authors response: Thanks to the Reviewer for his valuable comment. The authors have corrected the formatting errors throughout the entire manuscript.

8. Mistakes in the List of References has to be corrected;

a. Name of the journal missing: [16], [17]

b. et al.

Authors response: Thanks to the Reviewer for his valuable comment. To address this comment, the missing information has been added in the revised manuscript, and the format of the references has been improved and unified (please refer to pages 20-21).

9. Start from a Capital letter: [2]

Authors response: Thanks to the Reviewer for his comment. The authors have made the required change (please refer to page 21). 

Reviewer 3:

This work shows basic work of cost model for AM processes. The work shows an economical model developed with two case studies. The model is simple and it could be applied to different AM processes. AM processes are not as difficult as other manufacturing this is why the cost model is simple. I have one consideration to be taken in account.

1. Use Nowadays instead of Now a days

Authors response: Thanks to the Reviewer for his valid comment. To address the Reviewer’s concern, the phrase “Nowadays” has been replaced by “Now a days” in the revised manuscript (please refer to page 1). 

Reviewer 4:

1. The keywords should be improved, considering that the authors are actually proposing ABC method. Furthermore, what does FDM stand for?

Authors response: Thanks to the Reviewer for his important comment, and he is correct in noting that. Therefore, the keywords have been improved in the revised manuscript (please refer to page 1). Additionally, FDM stands for Fused Deposition Modeling, a commonly used additive manufacturing technique. In our revised work, we aimed to make the research more realistic, and plastic would no longer be used for real application but merely for experimental parts. For this reason, we limited ourselves to metal only because it is medically and practically suitable for the patient.

2. Usually, the acronyms defined in the abstract, should be re-defined in the manuscript (outside of the abstract)

Authors response: The Reviewer’s comment alerted us that the acronyms were not adequately used. Therefore, we ensured that acronyms defined in the abstract are re-defined in the revised manuscript (please refer to pages 1-2).

3. The way the abstract is organized suggest that the authors are suggesting ABC method, however, based on the introduction, it seems that there are existing ABC models suggested in the literature. Which brings these two questions: What are the novelties of this study? Is this paper proposing the ABC model for the first time?

Authors response: The Reviewer is correct in his conclusion. We would like to mention that the novelty of our research lies in implementing the cost model on a real case in the medical section. Additionally, we clearly state in our paper that it uses an ABC model based on the existing literature, as described on page 2.

4. The written quality of sections 2 and 3 are not acceptable and must be improved.

Authors response: Following the Reviewer’s recommendation, section 2 and 3 have been rewritten entirely, and their structure has been completely overhauled. We hope that they will meet the editors’ and reviewers’ expectation.

5. What does m refer to

---

## [Decision Letter · Decision Letter 1]

17 Mar 2024

ABC Model for Cost Estimation of Custom Implants by Additive Manufacturing

PONE-D-23-42855R1

Dear Dr. hameed,

We’re pleased to inform you that your manuscript has been judged scientifically suitable for publication and will be formally accepted for publication once it meets all outstanding technical requirements.

Kind regards,

Saliha Karadayi-Usta, PhD

Academic Editor

PLOS ONE

Additional Editor Comments (optional):

Reviewers' comments:

Reviewer's Responses to Questions

**Comments to the Author**

1. If the authors have adequately addressed your comments raised in a previous round of review and you feel that this manuscript is now acceptable for publication, you may indicate that here to bypass the “Comments to the Author” section, enter your conflict of interest statement in the “Confidential to Editor” section, and submit your "Accept" recommendation.

Reviewer #2: All comments have been addressed

Reviewer #3: (No Response)

2. Is the manuscript technically sound, and do the data support the conclusions?

Reviewer #2: Yes

Reviewer #3: (No Response)

3. Has the statistical analysis been performed appropriately and rigorously? 

Reviewer #2: N/A

Reviewer #3: (No Response)

4. Have the authors made all data underlying the findings in their manuscript fully available?

Reviewer #2: Yes

Reviewer #3: (No Response)

5. Is the manuscript presented in an intelligible fashion and written in standard English?

Reviewer #2: Yes

Reviewer #3: (No Response)

6. Review Comments to the Author

Reviewer #2: Reviewer’s comments on 11 03 2024 “ABC Model for Cost Estimation of Custom Implants by Additive Manufacturing”

Please refer further comments.

1. The novelty of this article which has been highlighted, do not prominent in the title, the abstract or in keywords. So re-write title, the abstract and keywords, removing unwanted definitions and clarifications.

• Rather writing ‘custom implants’ in the title, the authors can replace by customizing medical implants in the title and better little change the abstract and keywords accordingly. There, general or unnecessary explanations in the abstract can be removed.

• Nowadays, Additive Manufacturing (AM), often known as 3D printing, is a technology that is making significant progress in customizing medical implants for individuals.

• According to the authors, the novelties of this study enables the creation of complex geometries impossible with traditional production methods by implanting for unusually shaped areas, such as the skull or pelvis, can benefit greatly from this innovation.

• we are utilizing the MYSINT 100 for manufacturing custom implants, offering advantages such as high precision, design flexibility and efficient production. Its selective laser melting technology allows for intricate and patient-specific implant designs, contributing to better fit and enhanced functionality. The MYSINT 100's capabilities in processing various materials make it suitable for a range of implant applications, providing a reliable and customizable solution in AM for medical purposes.

• The authors aim to represent a common and diverse set of materials used in AM and to make the research more realistic. As, plastic would no longer be useful for real-world application but would serve merely as an experimental part thus limited to use only metal, as it is medically and practically suitable for the patient

2. in the Abstract

Computer-aided design (CAD) model – Change aided design to capital letters and write as Computer-Aided Design

3. Formatting errors have to be removed.

• Overhead Cost (justify the paragraph,

The overhead costs during the processing phase, administrative services, lighting, etc. are proportioned based on the setup and build time (). This rate (2) could deviate from preprocessing, as environments during these operations vary, hence, the overhead cost (2 ) is calculated as:

This ensures that overhead costs are accurately associated with the processing phase.

c. Post-processing Activity (check Tabs, font size and justification of the paragraph)

Post-processing in AM entails finishing and inspection, including steps like heat treating. Post-processing costs cover material, machinery, labor, energy, inspection, and overhead. The formula for the postprocessing cost is:

4. future work

In addition to incorporating visual aids such as charts or graphs to represent the cost breakdown more intuitively to make visualizations enhance the reader's understanding and facilitate a quicker grasp of complex information, you may develop a software in the future studies.

Reviewer #3: N/A

This work shows basic work of cost model for AM processes. The work shows an economical model developed with two case studies. The model is simple and it could be applied to different AM processes. AM processes are not as difficult as other manufacturing this is why the cost model is simple.

7. PLOS authors have the option to publish the peer review history of their article (what does this mean?). If published, this will include your full peer review and any attached files.

Reviewer #2: **Yes: **Kokila Madhuri Wijewickrama Abeykoon

Reviewer #3: No

---

## [Editor Report · Acceptance letter]

25 Apr 2024

PONE-D-23-42855R1 

PLOS ONE

Dear Dr. hameed, 

I'm pleased to inform you that your manuscript has been deemed suitable for publication in PLOS ONE. Congratulations! Your manuscript is now being handed over to our production team.

Kind regards, 

on behalf of

Assoc. Prof. Dr. Saliha Karadayi-Usta 

Academic Editor

PLOS ONE